# Cellular resolution circuit mapping with temporal-focused excitation of soma-targeted channelrhodopsin

Christopher A Baker[1]*, Yishai M Elyada[2†], Andres Parra[2], M McLean Bolton[1]*

[1]Disorders of Neural Circuit Function, Max Planck Florida Institute for Neuroscience, Jupiter, United States; [2]Functional Architecture of the Cerebral Cortex, Max Planck Florida Institute for Neuroscience, Jupiter, United States

**Abstract** We describe refinements in optogenetic methods for circuit mapping that enable measurements of functional synaptic connectivity with single-neuron resolution. By expanding a two-photon beam in the imaging plane using the temporal focusing method and restricting channelrhodopsin to the soma and proximal dendrites, we are able to reliably evoke action potentials in individual neurons, verify spike generation with GCaMP6s, and determine the presence or absence of synaptic connections with patch-clamp electrophysiological recording.

**\*For correspondence:**
christopher.baker@mpfi.org
(CAB); mclean.bolton@mpfi.org
(MMcLB)

**Present address:** [†]Department of Neurobiology, Institute of Life Sciences, Edmond and Lily Safra Center for Brain Sciences, Hebrew University of Jerusalem, Jerusalem, Israel

**Competing interests:** The authors declare that no competing interests exist.

## Introduction

The synaptic organization of individual neurons into circuits is the physiological basis for the interpretation of sensory input and production of behavioral responses. Understanding the precise patterns of connectivity among the distinct types of neurons that comprise neural circuits is critical for elucidating circuit function and ultimately requires methods that can map functional connectivity with single-cell resolution. Optical activation of single neurons using two-photon excitation of caged neurotransmitters or optogenetic probes such as channelrhodopsin (ChR2) provides a powerful approach for assessing the synaptic connections of single neurons. In particular, optogenetic mapping utilizing ChR2 and the rapidly expanding family of opsin variants have increased the flexibility and precision of mapping paradigms. Variations in the single-channel properties of the opsins can be exploited to generate rapid action potential trains or sustained depolarizations (*Mattis et al., 2012*), and new red-shifted variants have facilitated excitation deeper in tissue and have enabled simultaneous optical control of two distinct neuronal populations (*Klapoetke et al., 2014*; *Lin et al., 2013*; *Yizhar et al., 2011*). In addition, genetic restriction of opsin expression using transgenic mouse lines enhances the ability to activate and assess the connectivity of specific cell types.

Despite the great potential of optogenetics for mapping the synaptic connections of single neurons, there are multiple issues that have limited its effectiveness. First, two-photon activation of single neurons with ChR2 is complicated by its kinetics and low single-channel conductance. A diffraction-limited spot does not activate sufficient channels simultaneously to reliably bring neurons expressing ChR2 to action potential threshold. Several solutions have been implemented to address this. Rapid scanning of a diffraction-limited two-photon excitation spot across an opsin-expressing cell allows sufficient temporal integration to generate action potentials (*Packer et al., 2012*; *Prakash et al., 2012*; *Rickgauer and Tank, 2009*). Alternatively, scanless two-photon excitation by temporal focusing (*Oron et al., 2005*; *Zhu et al., 2005*) increases the number of simultaneously excited opsin molecules by expanding the beam in the imaging plane without sacrificing the optical sectioning of multiphoton microscopy (*Andrasfalvy et al., 2010*; *Losonczy et al., 2010*; *Papagiakoumou et al., 2010*; *Rickgauer et al., 2014*). The implementation of diffractive optical

**eLife digest** Nerve cells called neurons carry information around the body in the form of electrical impulses and pass signals to each another to form circuits that link different organs and tissues. Mapping out the neurons in the brain can reveal how different circuits contribute to an animal's behavior. Yet, because the brains of mammals contain millions of neurons, these circuits are often difficult to untangle.

One way to tease apart circuits of neurons uses a technique called optogenetics, which involves manipulating the genes inside neurons such that the cells produce a light-sensitive protein and respond to blasts of light. The aim is to activate a specific neuron and then see which other neurons are activated shortly afterwards, revealing a connected circuit. However, exposure to light can be imprecise. Also, the neurons in the brain are so densely packed that the nerve endings from neighboring neurons often overlap without actually being connected. This makes it unclear if activated neurons are truly part of the same circuit or simply bystanders reacting to the same nearby blast of light.

To overcome this limitation, Baker et al. developed a new optogenetic approach with two important features. First, the approach makes use of a light-sensitive protein called channelrhodopsin that had been modified to confine it to the cell body of each neuron and exclude it from the nerve endings. Second, pulses of laser light were specifically shaped to target only the cell body of an individual neuron. Baker et al. show that this new method can activate neurons inside slices of mouse brain without affecting the neighboring neurons. This allowed circuits of neurons to be mapped in fine detail.

This new optogenetic method is expected to shed light on the patterns of nerve signals that contribute to animal behavior. The approach may also be modified to use other light-sensitive proteins or investigate how neural circuits are altered in animal models of human disorders like autism and schizophrenia.

elements (*Fino and Yuste, 2011*; *Nikolenko et al., 2007*; *Packer and Yuste, 2011*) or spatial light modulators (*Dal Maschio et al., 2010*; *Nikolenko et al., 2008*; *Packer et al., 2012*, *2015*; *Papagiakoumou et al., 2010*, *2009*, *2008*) has also allowed for more complicated excitation profiles that encompass multiple spots around a cell, ensembles of neurons, or particular branches of dendritic trees.

While these approaches have made it possible to provide sufficient two-photon illumination to reliably drive action potentials, the ability to use optogenetic stimulation to selectively target single neurons remains challenging because the opsins are expressed throughout the dendritic and axon terminal fields, generating a potential confounding source of light-induced electrical responses. For example, it can be difficult to know whether a recorded electrophysiological event is due to stimulation of a presynaptic cell or 'direct' stimulation of a portion of the recorded cell's dendritic arbor. Although these possibilities can be distinguished by their kinetics, large amplitude direct responses may obscure simultaneous smaller synaptic events. This essentially leaves an indeterminate region of any circuit 'map' coinciding with the dendritic arbors of the recorded neuron, which can extend for 200 µm or more around the soma. In addition, the optogenetic approach could be compromised by unintended activation of fibers of passage or local axonal boutons, which are known to respond to temporal focusing of two-photon excitation (*Andrasfalvy et al., 2010*; *Losonczy et al., 2010*). Exclusion of opsins from axonal compartments has previously been achieved by fusing the opsin with targeting domains that bind to myosin Va motors necessary for transporting proteins into dendrites; one such motif from melanophilin is sufficient to exclude ChR2 from axons and enhance the resolution of neural circuit maps (*Lewis et al., 2009*). The resulting ChR2 distribution, however, remains throughout the dendritic tree and thus does not solve the problem of undesired direct activation of a neuron's processes while trying to stimulate other neurons in close proximity.

To overcome these limitations, we combined temporal focusing with spatial confinement of ChR2 expression to the neuronal cell body and proximal dendrites. Our alternative targeting approach took advantage of the Kv2.1 potassium channel, which has a particularly unique localization to

clusters at the neuronal soma and proximal dendrites (*Trimmer, 1991*) achieved through a 66-amino acid domain in its C-terminus (*Lim et al., 2000*) that drives its association with myosin IIb and specific post-Golgi transport vesicles destined for the somatic compartment (*Jensen et al., 2014*). Furthermore, this targeting signal is sufficient to alter ChR2 trafficking in retinal ganglion cells (*Wu et al., 2013*). We have employed this approach to target ChR2 to the soma and proximal dendrites of neurons in somatosensory cortex and added a nuclear fluorescent tag to identify ChR2-expressing neurons for targeting a two-photon temporal focused mapping beam. We combined this soma-targeted ChR2 with verification of successful action potential generation with a genetically encoded calcium indicator in our mapping protocol. Our approach allows robust and precise activation of neurons in brain slices for the construction of functional synaptic connectivity maps with single-cell resolution without loss of information about local connections in the region of the dendritic arbor of the recorded neuron or inadvertent activation of axons.

## Results

We stimulated neurons expressing soma-targeted ChR2 in acute slices of mouse cortex using scanless temporal focusing (TF), which has been used successfully for optogenetic stimulation at high axial resolution in scattering tissue samples (*Andrasfalvy et al., 2010*; *Losonczy et al., 2010*; *Papagiakoumou et al., 2010*, *2013*; *Rickgauer et al., 2014*). TF uses a diffraction grating (*Figure 1—figure supplement 1*) to separate the spectral components of a pulsed laser beam, resulting in a temporally broadened pulse that is inefficient at excitation except at the focal plane, where the components are recombined. This yields a volume of excitation in which the diameter in the x-y plane and the thickness in the axial plane are controlled independently (*Oron et al., 2012*, *2005*). We designed our excitation volume to approach the dimensions of a typical neuronal soma (*Figure 1—figure supplement 2*).

To restrict expression of ChR2 to the soma and proximal dendrites, we generated ChR2 fusion proteins by attaching a 65 amino acid motif from the Kv2.1 voltage-gated potassium channel to the carboxy terminus of ChR2-EYFP (*Lim et al., 2000*). Nontargeted ChR2-EYFP fluorescence was distributed throughout the processes of dissociated cortical neurons that had been filled with a fluorescent dye (*Figure 1A*). In contrast, targeted ChR2-EYFP-Kv2.1 was located primarily in the soma and excluded from the axon and distal dendrites (*Figure 1A*).

As a functional assay to compare the distribution of ChR2 in the soma and dendrites of targeted and non-targeted constructs, we used TF stimulation at intervals along the apical dendrite of patched layer II/III pyramidal neurons in acute coronal brain slices of somatosensory cortex from mice expressing opsins from viral constructs and recorded light activated currents in voltage clamp (*Figure 1B*). The stimulation power for these experiments was determined independently for each neuron to be the minimum that elicited a single action potential in 10 out of 10 trials when the TF spot was placed directly over the soma; this value was $0.92 \pm 0.24$ mW/$\mu$m$^2$ for targeted ChR2 and $1.65 \pm 0.26$ mW/$\mu$m$^2$ for the nontargeted version, owing to the increased sensitivity of the targeted construct (see below). Dendrites were followed throughout the depth of the tissue and areas were selected for stimulation such that the dendrite was planar throughout the extent of the 10 $\mu$m TF disc. The TF-stimulated current declined ~10-fold at 50 $\mu$m from the soma with the targeted construct, versus two fold for the non-targeted construct (*Figure 1C*; N=10 untargeted cells from four animals and N=13 targeted cells from six animals). The reduced dendritic ChR2 in the targeted construct opens up the possibility of using light stimulation with patch recording to identify synaptic currents that originate from nearby neurons. To illustrate this point, we compared the spatial distribution of direct currents recorded in Layer II/III pyramidal neurons in acute slices of somatosensory cortex expressing non-targeted and targeted constructs while sequentially stimulating points at 20 $\mu$m intervals over a 300 × 300 um grid surrounding the soma. Stimulation of cells expressing nontargeted ChR2-evoked currents up to 200 $\mu$m away from the neuronal soma, often delineating a pattern indicative of the dendritic arbor of the cell (*Figure 1D*). In contrast, cells expressing the targeted construct exhibited significant currents only when stimulated within the 25–50 $\mu$m immediately adjacent to the cell body. The lack of direct light-activated currents throughout much of the neuron's dendritic field makes it possible to visualize synaptic currents that would be evoked by ChR2 activation of nearby presynaptic neurons, even those that lie within the neuron's dendritic field.

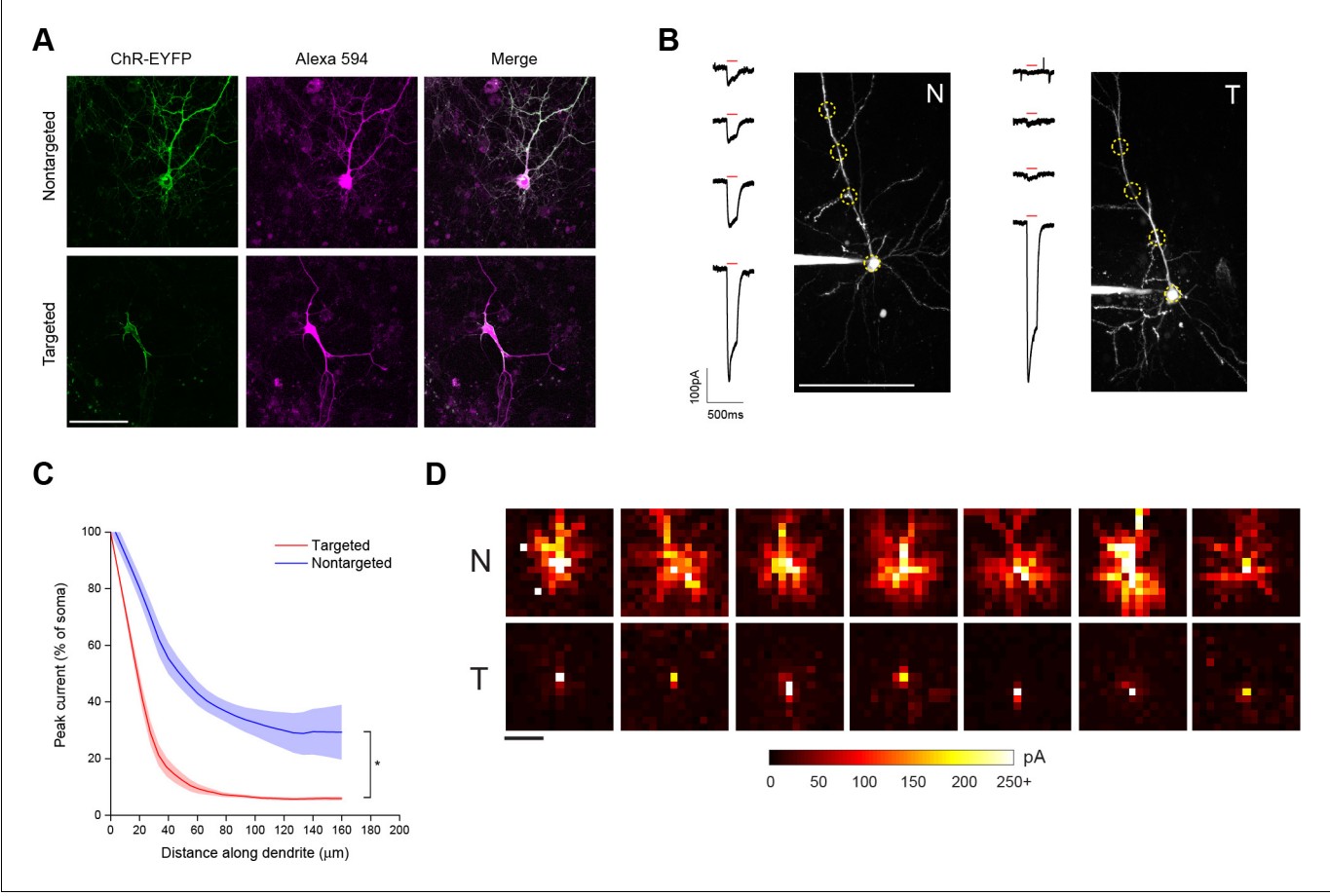

**Figure 1.** Characterization of soma-targeted channelrhodopsin (ChR2). (**A**) Fluorescence of ChR2-EYFP (green) in live dissociated cortical neurons that had been patched and filled with Alexa 594 dye (magenta) as visualized by two-photon microscopy. Scale bar = 100 μm. (**B**) Representative two-photon maximum intensity projections of Alexa 594 fluorescence and current responses to a single 150 ms TF stimulation pulse (red bar) for patched and dye-filled pyramidal cells in acute slices expressing nontargeted (N) or targeted (T) ChR2. Scale bar = 100 μm. (**C**) Mean (solid lines) ± s.e.m. (shaded regions) for current responses to TF stimulation measured at specific points along a single dendrite (N=10 nontargeted cells and 13 targeted cells). Fitting each curve with an exponential function demonstrated that the nontargeted and targeted datasets were significantly different (*p<0.05 by F-test). (**D**) Maps of direct currents in representative cells expressing ChR2 constructs when stimulating a 15 × 15 grid of locations. Each pixel in a map represents the direct current observed in the patched cell when that location in the slice was stimulated. Each position in a map was stimulated with the minimum power that evoked action potentials in 10/10 trials when stimulation was applied to the soma. Currents evoked by such powers ranged from 20 to over 250 pA, as indicated by the scale at bottom. Scale bar at lower left = 100 μm.

The following figure supplements are available for figure 1:

**Figure supplement 1.** Custom two-photon microscope setup for temporal focusing stimulation with simultaneous imaging.

**Figure supplement 2.** Orthogonal projections from a z-stack of a spin-coated layer of 200 nm fluorescent beads imaged with the temporal focusing beam and collected with a CCD camera.

**Figure supplement 3.** Characterization of targeted ChR2 in axonal processes.

In brain sections from mice injected with viruses encoding ChR2 constructs, examination of regions near the edge of the extent of virus-driven expression revealed markedly denser labeling of neuronal processes around a single-neuron expressing nontargeted ChR2 versus the area around several neurons expressing the targeted version (*Figure 1—figure supplement 3A*). We also saw processes reminiscent of axons at distances of several hundred microns away from the cell bodies of conventional ChR2-positive neurons. To functionally characterize the extent of axonal opsin

expression, we again patched layer II/III pyramidal neurons in acute coronal brain slices of somato-sensory cortex and recorded light-activated currents in voltage clamp while stimulating multiple positions along putative axonal processes, which were distinguished by their thin profile, absence of spines, and presence of occasional varicosities. Although the amount of detected current decreased rapidly for both constructs as TF stimulation was moved along the axon, noticeable currents could be detected at distances of 50 µm or greater from the soma of cells expressing nontargeted ChR2 (*Figure 1—figure supplement 3B*). In contrast, cells expressing targeted ChR2 demonstrated an 80% reduction of current when stimulating just 20 µm down the axon and significantly less current than cells expressing nontargeted ChR2 throughout its examined length (*Figure 1—figure supplement 3C*).

Interestingly, the targeted ChR2 also exhibited increased peak current amplitude in response to TF activation, presumably by concentrating channel density at the soma (*Figure 2A*; from 308.3 ± 43.5 pA in 12 nontargeted cells to 760.5 ± 146.0 pA in 12 targeted cells at 3.63 mW/µm$^2$; p=0.0101 for main effect of targeting by two-way repeated measures ANOVA), leading to an over three-fold reduction of the power required to reach action potential threshold (*Figure 2B*; from 2.75 ± 0.31 mW/µm$^2$ in 17 nontargeted cells to 0.88 ± 0.23 mW/µm$^2$ in 19 targeted cells; p=8.028 × 10$^{-5}$ by Mann-Whitney U Test). At the threshold stimulation power for each cell, the amount of evoked current was similar between targeted and untargeted constructs (261.06 ± 31.62 pA for untargeted ChR2 and 222.70 ± 27.21 pA for targeted ChR2 p=0.3677 by two sample t-test), consistent with the observation that the targeting modification had no effect on rheobase, other intrinsic physiological parameters, or action potential properties in response to current injections (*Figure 2—figure supplement 1* and *Table 1*).

The ability to trigger action potentials at lower incident power, coupled with the somatic restriction of the targeted construct, should provide enhanced spatial and temporal resolution for mapping neuronal circuits. We measured the spatial resolution of action potential generation by moving the TF spot to different lateral and axial locations relative to a patched neuron expressing targeted ChR2 and examining the proportion of 10 trials that resulted in an action potential when using the threshold stimulation intensity. The full-width at half maximum of these measurements was 11.1 µm laterally and 23.3 µm axially (N=10 cells from 7 animals; *Figure 2C and D*). Similar measurements using nontargeted ChR2 demonstrated resolution of 19.6 µm laterally and 36.2 µm axially (N=13 cells from 5 animals; *Figure 2C and D*); both curve fits were significantly different from the targeted ChR2 versions (p<0.05 by F-test). In terms of temporal precision, the mean latency from light onset to generation of a single action potential in the above experiments was 38.98 ± 17.33 ms; the average jitter (defined as the standard deviation of the latency across the ten trials for a given cell) was 6.8 ± 2.1 ms. Increasing stimulation power beyond threshold provoked additional action potentials and eventually shortened the latency to first spike to 9.3 ± 1.8 ms (*Figure 2—figure supplement 2*), consistent with reported values for stimulating cortical neurons with sculpted light (*Papagiakoumou et al., 2010*). Targeted ChR2-expressing cells exhibited shorter action potential latencies than nontargeted ChR2 cells at equivalent stimulation intensity, consistent with latency being power-dependent and the earlier observation that targeted ChR2 is more sensitive. Although we concentrated efforts on longer stimulation pulses to increase the number of spikes and facilitate calcium imaging (see below), we verified the performance of targeted ChR2 under a shorter stimulation regimen. Temporal focusing pulses of 32 ms were effective at generating action potentials in opsin-expressing cells, and targeted ChR2 exhibited greatly enhanced photocurrents and lower power thresholds under these conditions (*Figure 2—figure supplement 3*). Moreover, the ability to reduce latency to less than 10 ms with increased power (*Figure 2—figure supplement 2*) suggests that even shorter stimulation pulses may still be effective. Therefore, the combination of temporal focusing and soma-targeted ChR2 expression yields a highly reliable and spatially precise means to stimulate action potentials with somatic illumination.

To demonstrate the general utility of these techniques for mapping synaptic connections, we performed connectivity experiments in acute somatosensory cortical slices. We generated a bicistronic adeno-associated virus (AAV) construct encoding targeted ChR2-Kv2.1 followed by a P2A ribosomal skipping sequence and a histone 2B-mRuby2 fusion protein to fluorescently label neuronal nuclei to identify cells for TF stimulation. We patched a cell in layer II/III, and then stimulated surrounding cells that had been identified by nuclear mRuby2 fluorescence. Action potential firing in a light-stimulated neuron was verified by an increase in the fluorescence signal measured with the genetically encoded

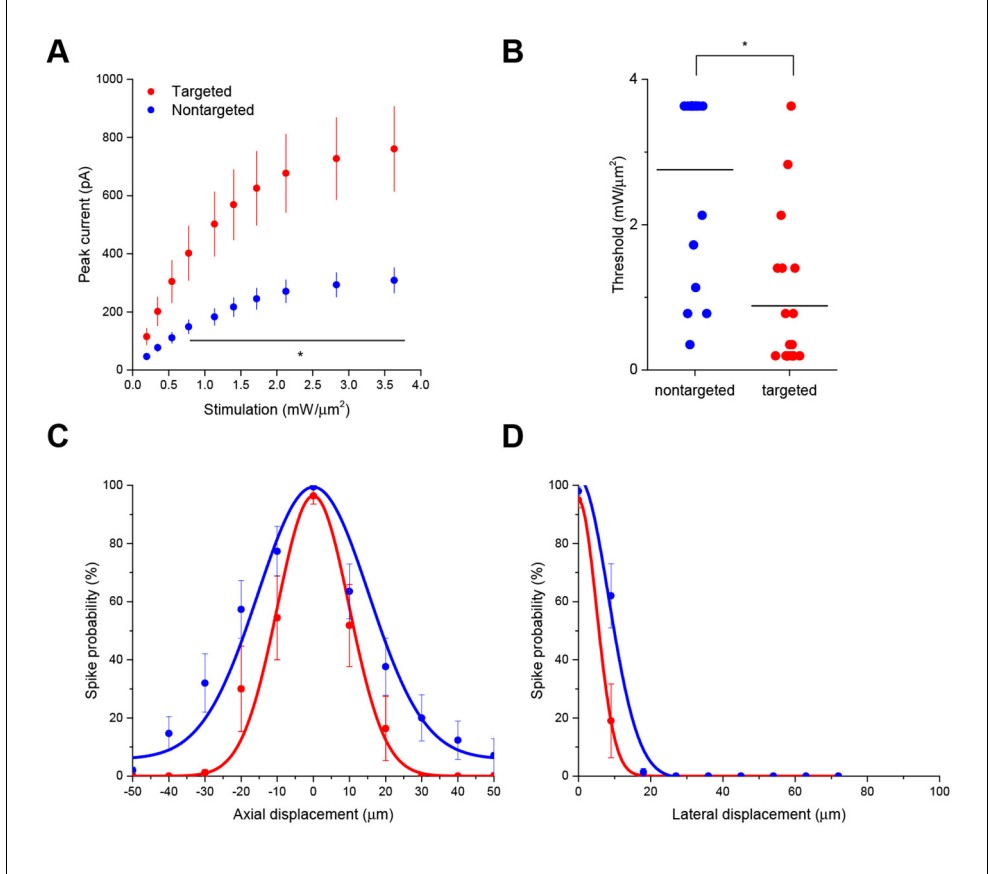

**Figure 2.** Enhanced sensitivity and resolution with soma-targeted ChR2. (**A**) Evoked currents in acute slices in response to 150 ms TF stimulation of increasing intensity; targeted ChR2 exhibited significantly greater current (two-way repeated measures ANOVA, p=0.0101 for main effect of targeting, p=0.0056 for targeting × power interaction, *p<0.05 Holm-Bonferroni post hoc test). (**B**) Minimum TF power required to elicit a single action potential. Each point indicates a single cell; cells expressing targeted ChR2 had a much lower threshold (*p=8.028 × 10$^{-5}$, Mann-Whitney U Test). (**C–D**) Probability of evoking spikes per 10 trials in cells expressing targeted (red) or nontargeted (blue) ChR2 when stimulating the soma and at laterally and axially displaced locations.

The following figure supplements are available for figure 2:

**Figure supplement 1.** Action potentials triggered in neurons expressing nontargeted (N) or targeted (T) ChR2 in response to stepwise current injections ranging from −100 to 110 pA in 30 pA steps.

**Figure supplement 2.** Action potential latency as a function of stimulation power in cells expressing targeted (red) or nontargeted (blue) ChR2.

**Figure supplement 3.** Latency to first action potential with targeted and nontargeted ChR2.

---

calcium sensor GCaMP6s (*Chen et al., 2013*) expressed from a separate AAV construct. For these experiments, we used an average stimulation power (2.29 ± 0.55 mW/μm$^2$) that, when combined with a 150 ms stimulation pulse, frequently led to trains of 2–4 action potentials in the patched cells and facilitated detection of larger calcium transients. We used an imaging power and dwell time that did not lead to action potential generation in any cells recorded, instead causing an average inward current of 29.5 ± 9.5 pA (N = 7 cells from 5 animals)—well below the average rheobase. The ability to identify a ChR2-Kv2.1-expressing neuron for TF stimulation by the presence of a fluorescent nuclear label and verify that the neuron has in fact been stimulated successfully by detecting calcium

**Table 1.** Intrinsic electrophysiological properties of neurons expressing normal or targeted channel rhodopsin constructs. Rheobase measurements were made in response to current injections. All measurements are mean ± s.e.m. No significant differences as an effect of targeting were found (Mann-Whitney U Tests, all p>0.05).

| | Nontargeted | Targeted | p |
|---|---|---|---|
| Input resistance (MΩ) | 129.16 ± 8.91 (N = 11) | 148.36 ± 17.76 (N = 18) | 0.9462 |
| Capacitance (pF) | 125.25 ± 16.99 (N = 11) | 89.85 ± 7.66 (N = 18) | 0.0887 |
| Resting potential (mV) | −72.71 ± 3.86 (N = 7) | −76.87 ± 1.87 (N = 8) | 0.3532 |
| Rheobase (pA) | 160.00 ± 7.69 (N = 12) | 147.39 ± 12.92 (N = 23) | 0.5357 |
| Spikes at 1.5× rheobase | 7.18 ± 0.52 (N = 12) | 7.56 ± 0.83 (N = 23) | 0.8377 |

transients with GCaMP is a key advantage for the execution and interpretation of mapping experiments.

Excitatory synaptic connections were identified by the presence of GCaMP fluorescence increase in only the TF stimulated neuron and a reproducible inward current with appropriate synaptic delay, kinetics and reversal potential in the patched cell (see Materials and methods). To separate spontaneous currents occurring during the stimulation epoch from bona fide synaptic events, we also required occurrence of synaptic responses on multiple stimulus repetitions and with a post-stimulus onset jitter of less than 14 ms. A lack of connection was defined as failure to reach these criteria following the presence of a GCaMP6s response to light stimulation in a potential presynaptic neuron.

In a representative experiment, we identified 43 nuclear-labeled cells in a single axial plane (*Figure 3A*), of which 35 cells yielded calcium transients in response to TF stimulation (*Figure 3B*). Three of these 35 cells elicited reproducible postsynaptic responses in the recorded neuron when photostimulated (*Figure 3C*), exhibiting multiple currents consistent with the production of trains of presynaptic action potentials. Every connection detected was associated with an unequivocal calcium transient in only the stimulated neuron (*Figure 3B*). In repeated experiments (seven neurons from five animals), another example of which is shown in *Figure 3—figure supplement 1*, the average rate of detecting a calcium response to TF stimulation of a cell expressing the ChR2-Kv-P2A-H2B-mRuby2 bicistronic construct was 80.00 ± 2.60%. Given that the GCaMP6s is expressed by a separate AAV construct and coinfection is not necessarily 100%, the 'nonresponsive' cells may express insufficient GCaMP for detection of single action potentials. Indeed, we identified three reproducible postsynaptic events (out of 316 presynaptic stimulations) that could not be correlated with a calcium transient, perhaps again due to lower GCaMP expression in those neurons. Overall, the average connectivity rate was 10.27 ± 2.60% (27 connections out of 252 cells showing a calcium transient to photostimulation), which was not significantly different from our own results using paired recording (7 connections out of 115 cells; Fisher's exact test, p=0.18).

For comparison, we executed similar GCaMP6s-monitored mapping experiments (three neurons from two animals) using nontargeted ChR2 and the same average stimulation power (*Figure 4*). Although we were able to detect putative synaptic connections, they were frequently coincident with direct current responses indicative of stimulation of the patched cell's dendritic arbor (*Figure 4C*). Across the experiments, such direct responses occurred during stimulation of 28.75 ± 4.46% of the target cells within a single field of view. These direct responses were often large in amplitude and could easily obfuscate much smaller synaptic events associated with bona fide connections. Moreover, we also observed reproducible calcium transients in off-target cells when other target cells were being stimulated (note events in *Figure 4B* that lie off of the diagonal). These off-target responses occurred with a probability (the number of events divided by the number of target cells) of 15.17 ± 1.79% and likely result from unintentional stimulation of a ChR2-containing sensitive dendrite of one cell while intending to activate the soma of a separate cell. Consistent with this interpretation, these off-target calcium events were observed at a significantly lower frequency in experiments with the targeted opsin (5.42 ± 1.39%, p<0.01 by two sample t-test). Moreover, the average distance between a cell exhibiting an off-target calcium transient and the intended target cell was greater with nontargeted ChR2 (40.6 ± 6.4 µm versus 18.9 ± 2.4 µm; p<0.01 by two sample

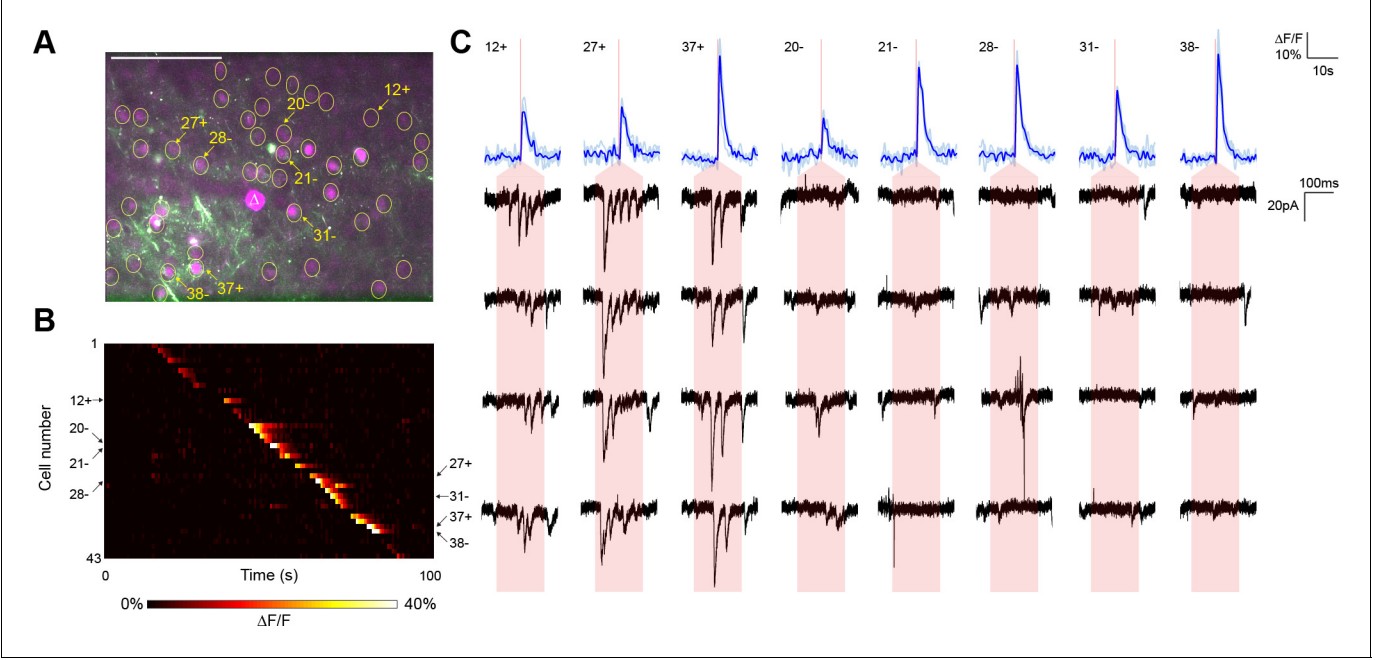

**Figure 3.** Typical mapping experiment with action potential validation by GCaMP6s. (**A**) In acute slices expressing GCaMP6s (green) and targeted ChR2-P2A-H2B-mRuby2 (magenta), cells were identified for photostimulation based on nuclear mRuby2 fluorescence (yellow circles). A cell was patched and dye-filled (triangle) and assessed for postsynaptic currents as surrounding cells were stimulated (1.40 mW/μm$^2$ incident power) and GCaMP fluorescence was simultaneously recorded. Scale bar = 100 μm. (**B**) Changes in GCaMP fluorescence over the experimental timecourse for each cell identified in (**A**); each cell was stimulated sequentially with ~2 s between cells. Signals are the average of four trials. In this experiment, 35 out of 43 cells yielded calcium transients in response to optical stimulation; overall the average probability of detecting an induced calcium response was 80%. (**C**) Calcium (blue) responses for a subset of cells reacting to TF stimulation; four trials are overlaid in light blue, with the average in dark blue. Red lines in the calcium traces indicate the onset of stimulation for each cell. The recorded currents for the four trials (black) in the patched and putatively postsynaptic cell are shown expanded below each calcium trace; the shaded red region indicates the 150 ms stimulation epoch. Three cells showed a calcium response to TF stimulation and triggered postsynaptic currents (fast onset kinetics but delayed with respect to the TF stimulation) in the patched cell. Five representative cells showing calcium transients but no reproducible postsynaptic currents are also shown. For the displayed three connected cells and the five unconnected cells, the cell numbers, spatial locations and full calcium traces are marked with plus signs and minus signs, respectively, in panels (**A**) and (**B**). Across multiple experiments, the average connection probability was 10% (27 connections out of 252 responsive cells).

The following figure supplement is available for figure 3:

**Figure supplement 1.** Second typical mapping experiment with action potential validation by GCaMP6s.

t-test), further suggesting that cells expressing nontargeted ChR2 were firing action potentials in response to unintended stimulation of distal dendrites. Our combination of TF and restricted ChR2 thus facilitated the mapping of local circuitry within the 300 μm surrounding a neuron without confounding signals from its dendritic arbor and with a higher throughput than that achieved with electrophysiological techniques alone.

## Discussion

This study demonstrates that combining temporal focusing for two-photon activation of ChR2 with restriction of ChR2 expression to the soma and proximal dendrites of neurons yields a reliable method for evaluating synaptic connectivity with single-neuron resolution. The spatially restricted ChR2 expression we describe allows unmasking of synaptic connections from neurons whose somata lie close to the dendrites of the postsynaptic cell and would have been occluded by direct activation of ChR2 on the dendrite. In addition, depletion of ChR2 from axons prevents inadvertent depolarization of boutons or fibers of passage that could compromise attempts to identify the source of a

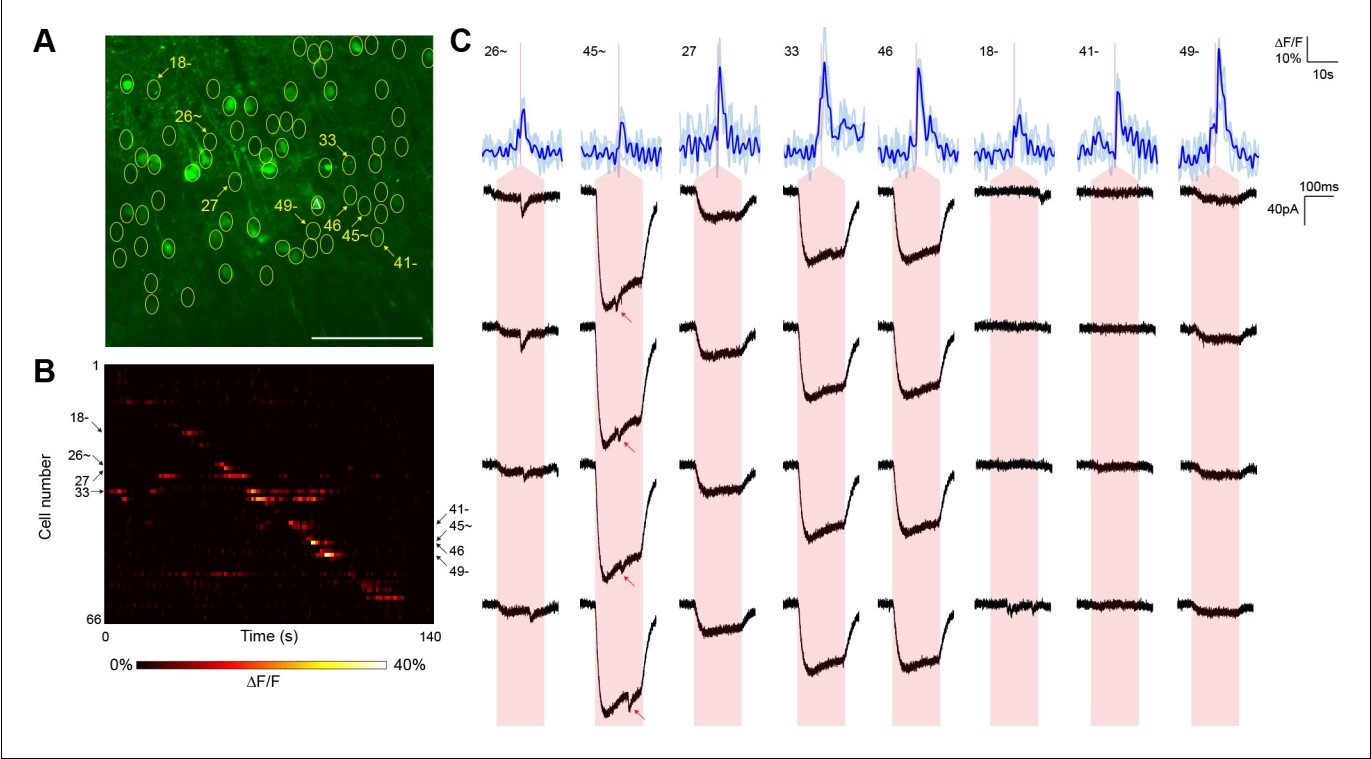

**Figure 4.** Typical mapping experiment using nontargeted ChR2. (**A**) In acute slices expressing GCaMP6s (green) and conventional ChR2-mRuby2, cells were identified for photostimulation based on responsiveness to widefield ChR excitation (yellow circles). A cell was patched and dye-filled (triangle) and assessed for postsynaptic currents as surrounding cells were stimulated (2.13 mW/μm² incident power) and GCaMP fluorescence was simultaneously recorded. Scale bar = 100 μm. (**B**) Changes in GCaMP fluorescence over the experimental timecourse for each cell identified in (**A**); each cell was stimulated sequentially as described in *Figure 3*. Signals are the average of four trials. (**C**) Calcium (blue) responses for a subset of cells reacting to TF stimulation; four trials are overlaid in light blue, with the average in dark blue. Red lines in the calcium traces indicate the onset of stimulation for each cell. The recorded currents for the four trials (black) in the patched and putatively postsynaptic cell are shown expanded below each calcium trace; the shaded red region indicates the 150 ms stimulation epoch. Several types of events were observed: one cell showed a synaptic event coincident with a low amplitude current from direct stimulation of the dendritic arbor of the patched cell, and one cell showed what might be a synaptic event buried within another direct stimulation current of large amplitude (red arrows). Three other cells with a calcium response to TF stimulation and coincident direct stimulation of the patched cell's dendritic arbor are also shown, along with three cells with calcium transients but no detectable currents in the patched cell. The cell numbers, spatial locations and full calcium traces corresponding to these events are marked in panels (**A**) and (**B**).

synaptic event. ChR2 harboring the Kv2.1 localization motif also showed enhanced sensitivity, which is of particular use in situations where excitation power is at a premium. Together these features of the soma-restricted construct significantly enhance the ability to map synaptic connections with single-cell resolution.

The development of spatially-restricted optogenetic constructs is probably one of the most effective means for achieving single-cell resolution in local circuit mapping experiments. This restriction requires the identification of a relatively small compartment near the neuronal soma characterized by selective expression of particular proteins with known motifs responsible for their localization. The distribution of particular voltage-gated potassium channels defines such a subcellular domain along the soma and proximal dendrites (*Trimmer, 2015*), and the sufficiency of a short Kv2.1 C-terminal sequence for driving heterologous proteins to this region (*Lim et al., 2000*) made this strategy ideal for restricting ChR2 expression. The only obvious alternative is restriction to the axon initial segment (AIS) mediated by an ankyrin G binding motif present in voltage-gated sodium channels (*Garrido et al., 2003*). Although incorporating this motif into ChR2 drives its localization to the AIS (*Grubb and Burrone, 2010*; *Wu et al., 2013*), the resulting construct does not support light-generated action potentials under physiological conditions (*Grubb and Burrone, 2010*). Moreover, AIS-

targeted ChR2 alters the intrinsic firing properties of retinal neurons, presumably by displacing endogenous voltage-gated sodium channels whose subcellular localization is also dependent on ankyrin G binding (*Zhang et al., 2015*). Therefore, the Kv2.1 targeting strategy currently remains the optimal means of concentrating ChR2 such that action potentials will not be generated by TF stimulation of dendrites >20 µm away from the soma. Indeed, this approach has been successfully used with opsins to artificially generate center-surround receptive fields in retinal ganglion cells (*Wu et al., 2013*). The *Wu et al. (2013)* study showed functional restriction of opsins to the soma and proximal dendrites by examining receptive field responses to wide (200 µm) bars of one-photon light. We have now measured the somatic targeting at a finer scale, demonstrated two-photon excitation of targeted ChR2 that is enhanced relative to normal ChR2 under the same conditions, and established the utility of targeted opsins for enhancing the resolution of local connectivity maps of neural circuits.

The soma-targeted ChR2 could be combined with many of the techniques for two-photon optical control of neuronal activity previously developed for caged neurotransmitters or optogenetic probes. The relative merit of the optical activation method depends on the axial resolution, temporal precision required and the number of neurons to be stimulated simultaneously in a given experimental paradigm. For example, rapid scanning of a diffraction-limited two-photon excitation spot in a pattern on the soma can generate action potentials (*Packer et al., 2012*; *Prakash et al., 2012*; *Rickgauer and Tank, 2009*) and combination with spatial light modulators allows for simultaneous excitation of neuronal ensembles (*Packer et al., 2012*, *2015*). The temporal resolution of this rapid scanning approach, however, is limited by the time required to scan along the cell body. Scanless activation of untargeted ChR in brain slices using TF, alone or in combination with spatial light modulators, is capable of generating action potentials in hippocampal pyramidal neurons within 1–3 ms of light onset (*Andrasfalvy et al., 2010*) and in cortical neurons in less than 10 ms (*Papagiakoumou et al., 2010*). We did not take full advantage of the temporal precision capability of TF to fire action potentials in our current study, instead focusing on a screening method that would identify connections without optimizing the amount of power that would fire each potential presynaptic neuron with minimal latency. We therefore chose longer pulses at a power sufficient to fire most neurons and generate trains of action potentials, which would elicit stronger signals with calcium indicators. For experiments requiring temporal precision, the minimization of action potential latency requires optimization of excitation area and laser power (*Papagiakoumou et al., 2008*). Because power is also the limiting factor in extending the area of two-photon activation to large numbers of neurons with spatial light modulators, the reduced power required to bring soma-targeted opsins to threshold would be an asset to such experiments.

The optimal performance of our method relies on sufficient co-expression of an opsin, a marker for opsin expression, and a genetically encoded calcium sensor. For GCaMP6s and similar sensors, there appears to be a balance between sufficient expression to detect single action potentials and excessive levels that lead to a lack of responsiveness (*Chen et al., 2013*; *Packer et al., 2015*; *Tian et al., 2009*). Targeting this expression window can be complicated, particularly when simultaneously trying to achieve high opsin levels using a separate AAV construct. Indeed, heterogeneity in GCaMP expression levels between neurons with viral infection may be partially responsible for the 20% of neurons in which we did not detect a change in GCaMP fluorescence with TF stimulation. We have also explored the co-expression of GCaMP6s as the second member of a single P2A-mediated bicistronic construct, but the GCaMP6s expression level was lower than desired, resulting in inadequate detection of action potentials (data not shown). Future development of these techniques will therefore benefit from bypassing viral systems and instead expressing the calcium sensor or the opsin in transgenic animals. The enhanced sensitivity of the targeted ChR2 should alleviate the concern that there would be sufficient opsin expression from a single genetic locus for activation of neurons with two-photon excitation.

There are several opportunities for future extension of this method. The somatic targeting approach could also be exploited in the context of other opsins such as C1V1, which has recently been leveraged for two-photon stimulation of neural circuits in acute slices and in vivo (*Packer et al., 2012*, *2015*; *Prakash et al., 2012*; *Rickgauer et al., 2014*). Other molecules which may be good candidates for somatic targeting include the red-shifted ReaChR for its sensitivity (*Lin et al., 2013*) and Chronos for its rapid temporal characteristics (*Klapoetke et al., 2014*). Moreover, the somatic restriction of membrane-bound genetically encoded voltage sensors could

dramatically reduce background associated with the neuropil and facilitate an all-optical version of our current method. Because the Kv2.1 targeting sequence is unable to restrict the distribution of the single-pass membrane protein CD8, however, there may be certain structural constraints on the effectiveness of the motif (*Lim et al., 2000*). We suspect that at least for the opsins, which are all seven transmembrane domain proteins likely to have a similar structure, the Kv2.1 motif will be sufficient to achieve somatic localization.

In summary, our combination of TF and soma-restricted ChR2 enables functional connectivity mapping and is straightforward and easy to implement with standard two-photon microscopes. Furthermore, these techniques could also be used in vivo, where the enhanced resolution of the targeted ChR2 makes it especially attractive for selective cell stimulation in behavioral paradigms. These enhancements to probing brain microcircuitry through optical stimulation promise to reveal much about nervous system function and how it might be modified by experience and perturbed in animal models of neurologic or psychiatric disease.

## Materials and methods

### Construct generation

The vector pAAV-hSyn-hChR2(H134R)-EYFP (Addgene plasmid 26973) served as the backbone for generating certain modified constructs. The 'proximal restriction and clustering signal' (*Lim et al., 2000*) of the Kv2.1 voltage-gated potassium channel (QSQPILNTKEMAPQSKPPEELEMSSMPSPVA PLPARTEGVIDMRSMSSIDSFISCATDFPEATRF), codon optimized for mouse, was generated by automated gene synthesis (Integrated DNA Technologies, Coralville, IA) and amplified by PCR using primers CGGCATGGACGAGCTGTACAAGCAGTCCCAGCCTATTCTGAAC and TGATATCGAATTC TTACTTAAACCGCGTAGCCTCTGG. The resulting product was inserted into the *Bsr*GI site at the C-terminus of the ChR2-EYFP fusion protein sequence using the Gibson Assembly kit (New England Biolabs, Ipswich, MA). For mapping experiments using nontargeted hChR2 with coincident visualization of GCaMP6s, the EYFP-coding sequence between the *Psh*AI and *Bsr*GI sites was replaced with the sequence of mRuby2.

To better visualize cells for stimulation during mapping experiments, we generated a bicistronic AAV construct consisting of hChR2 followed immediately by the Kv2.1 targeting sequence, a P2A ribosomal skipping sequence, and a histone 2B-mRuby2 fusion protein. The Kv2.1 sequence was amplified with primers ATCGAGGTCGAGACTCTCGTCGAAGACGAAGCCGAGGCCGGAGCCG TGCCAGCGGCCGCCACCCAGTCCCAGCCTATTCT and ACGTCTCCTGCTTGCTTTAACAGAGA GAAGTTCGTGGCTCCGGATCCAAACCGCGTAGCCTCTGG, histone 2B was amplified from Addgene plasmid 11680 with primers CTGTTAAAGCAAGCAGGAGACGTGGAAGAAAACCCCGGTCC TGGTTCTATGCCAGAGCCAGCGAAG and TCGGCTTCGTCTTCGACGAGCATGGTGGCGACCGG TG, and mRuby2 was amplified from Addgene plasmid 40260 with primers CTCGTCGAAGAC GAAGCCGAGGCCGGAGCCGTGCCAGCGGCCGCCACCATGGTGTCTAAGGGCGAA and GA TAAGCTTGATATCGTTACTTGTACAGCTCG. The amplified products were incorporated into an *Eco*RI-*Psh*AI fragment from Addgene plasmid 26973 by Gibson Assembly; the P2A sequence was reconstituted from the primers during the assembly process.

All plasmids were propagated in Stbl3 cells (Life Technologies, Grand Island, NY) and made into recombinant adeno-associated viral (AAV) particles of serotype 1 by the Penn Vector Core (Philadelphia, PA).

### Animals and viral injections

All animal work was conducted according to the Guide for the Care and Use of Laboratory Animals from National Institute of Health. C57BL6/J mice were maintained on a 12 hr light-dark cycle with ad libitum access to food and water. Under isofluorane anesthesia, P21-P25 mice were injected with AAV particles (800 nl containing ~$1 \times 10^{13}$ genome copy units/ml) at 9.2 nl per 10 s through a pipette positioned 500 µm beneath the surface of the somatosensory cortex and attached to a Nanoject II microinjector. For mapping experiments with nuclear labeling and verification of action potential generation by calcium influx, full strength AAV1 encoding ChR2-Kv2.1-P2A-H2B-mRuby2 ($5 \times 10^{12}$ units/ml) was coinjected with a diluted concentration of AAV1 GCaMP6s (final concentration = $1.4 \times 10^{12}$ units/ml); diluted GCaMP6s reportedly results in nuclear exclusion of the

calcium indicator and lower toxicity (*Packer et al., 2015*). For all data reported here, animals receiving targeted or untargeted ChR2 viruses were all examined between 3 and 4 weeks after injection. Somatic restriction was also observed in animals 5 weeks after injection.

### Dissociated cortical neurons

Neurons were prepared by dissecting the corticies from postnatal day 1 animals and digesting for 30 min at 37°C in Earle's Balanced Salt Solution supplemented with 1.5 mM $MgSO_4$, 1 mM $CaCl_2$, and 8.3 units/ml papain (Worthington Biochemical, Lakewood, NJ) under 95%$O_2$/5%$CO_2$. Just prior to plating, cells were transfected with 1 µg DNA per $2.5 \times 10^6$ cells using the Amaxa nucleofector system. Neurons were then seeded onto laminin-coated coverslips containing a feeder layer of astrocytes prepared as previously described (*McCarthy and de Vellis, 1980*). Cells were maintained in neuronal growth media (Neurobasal (Life Technologies) supplemented with 5 µg/ml insulin, 110 µg/ml sodium pyruvate, 100 units/ml penicillin, 100 µg/ml streptomycin, 40 ng/ml thyroxine, 292 µg/ml L-glutamine, 5 µg/ml N-acetyl-L-cysteine, 100 µg/ml BSA, 100 µg/ml transferrin, 16 µg/ml putrescine, 60 ng/ml progesterone, 40 ng/ml sodium selenite, 50 ng/ml BDNF, and 5 ng/ml forskolin) for ~10 days before use in experiments.

### Slice preparation and electrophysiology

Three weeks after viral injection, 300–400 µm slices were prepared in ice-cold cutting solution containing (in mM): 124 choline chloride, 26 $NaHCO_3$, 2.5 KCl, 3.3 $MgCl_2$, 1.2 $NaH_2PO_4$, 1 glucose and 0.5 $CaCl_2$. After cutting, slices were allowed to recover for 30 min at 32°C in artificial CSF containing (in mM): 124 NaCl, 26 $NaHCO_3$, 3 KCl, 1.25 $NaH_2PO_4$, 20 glucose, 1 $MgCl_2$, 2 $CaCl_2$, 5 sodium ascorbate, 3 sodium pyruvate and 2 thiourea. All solutions were maintained under constant 95%$O_2$/5%$CO_2$. Whole-cell recordings were made through 4-7 MΩ pipettes, filled with intracellular solution containing (in mM): 145 potassium gluconate, 5 NaCl, 10 HEPES, 0.5 EGTA, 4 MgATP, 0.3 NaGTP, 0.02 Alexa Fluor 488 or 594 hydrazide. In some experiments, the intracellular solution was supplemented with 0.2% biocytin. All recordings were collected using Axon Multiclamp 700B amplifiers and Digidata 1440A digitizers (Molecular Devices, Sunnyvale, CA) at 10 kHz, controlled by Clampex software.

### Optical setup

Our customized optical setup was based on a AxioImager Z1 platform (Zeiss, Thornwood, NY) fitted with an Ultima dual path scan head containing two pairs of galvanometric mirrors (Bruker, Middleton, WI) for separate control of imaging and stimulating optical pathways (*Figure 1—figure supplement 1*). The imaging pathway used a Chameleon Ti:sapphire laser tuned to 920 nm (Coherent, Santa Clara, CA). The stimulation pathway employed a MaiTai DeepSee Ti:sapphire laser tuned to 880 nm (SpectraPhysics, Santa Clara, CA). The spectral broadening necessary for temporal focusing was achieved on this path by placing a 300 line/mm diffraction grating (Thorlabs, Newton, NJ) approximately 1 m away from the galvanometric mirrors; the laser spot on the grating was imaged onto the plane between the mirrors using a 500 mm focal length lens (Thorlabs) placed 500 mm away from the grating along the path of the first diffraction order from the grating. The stimulation laser was recombined with the imaging laser in the scan head using a 900 nm long-pass dichroic mirror (Chroma, Bellows Falls, VT). The power applied by either laser was controlled with a separate Pockels cell (Conoptics, Danbury, CT).

### Confocal microscopy

Three weeks after viral injection, animals were transcardially perfused with saline followed by 4% paraformaldehyde. Brains were removed, cryoprotected in 30% sucrose, and sectioned at 50 µm intervals on a sliding microtome. Sections were stained for ChR2-EYFP fusion proteins with a rabbit antibody raised against GFP followed by an Alexa 488-labeled secondary antibody (Thermo Fisher Scientific, Waltham, MA). Sections were mounted in Prolong Gold (Thermo Fisher Scientific) and images were collected on a Zeiss 780 confocal microscope using 10× air- and 40× oil-immersion objectives. Acquisition settings for laser power and PMT voltages were kept constant between nontargeted and targeted ChR2 samples.

## Stimulation and resolution studies

Temporal focusing stimulation was controlled by Prairie View 5.0 software (Bruker). Standard stimulation protocols used 150 ms pulses, except as noted in the text. Stimulation powers were measured after the objective, and ranged from 15 to 285 mW; assuming our spot size to be at least 10 μm in diameter, we are using powers of no greater than 0.2 to 3.6 mW/μm$^2$. This calculation assumes a completely flat excitation profile; in reality, there is some spread in the axial dimension and thus, the power density is probably even lower. Cells filled with fluorescent Alexa dyes were visualized by two-photon microscopy and recorded in whole-cell voltage or current clamp as the temporal focusing spot was moved to different axial planes, different cell bodies or particular spots along dendrites. For each dendrite examined, the temporal focusing-induced current was plotted as a function of distance from the cell soma; the resulting curves were averaged for each construct to generate the traces shown in *Figure 1C*. All responses in *Figure 1* were verified to exhibit kinetics associated with direct stimulation of the patched cells: that is, exhibiting onset and offset precisely locked to the start and end of the optical stimulation. Stimulation powers were chosen separately for each cell by identifying the minimum power that evoked action potentials continually across ten trials. Resolution of spike generation was measured by moving the stimulation spot at 9 μm intervals laterally or 10 μm axially above or below the cell soma and recording the proportion of trials out of 10 leading to an action potential. Data were collected from cells until a minimum of ten cells was obtained for following ChR2 expression along dendrites; given a standard deviation of ~11% for these measurements, our sample size should be able to detect a 15% difference between groups with a power of 0.8. Data for other measures were collected in parallel, and no data were ever excluded even if a recording failed prior to collecting the dendrite expression results.

## Calcium imaging and connection analysis

Images of 512 × 512 pixels were simultaneously collected at 1.5 Hz using a raster scanning galvonometric system. Each stimulation epoch was timed to coincide with the onset of every other imaging frame, that is, the first spot was stimulated at the onset of the second frame, the second spot was stimulated 1.26 s later at the onset of frame 4, and so on. In this manner, a calcium peak in any frame could be associated with stimulation of a particular point in the field. During each stimulation epoch, microscope PMTs were protected from saturating fluorescence signals by Uniblitz shutters (Vincent Associates, Rochester, NY). Because this resulted in a lack of signal in the first 150 ms of every other frame as the top portion of the microscope field was scanned by the imaging laser, all stimulation points and calcium data collection were confined to a ~200 × 300 μm region of the microscope field. First-pass regions of interest (ROIs) were based on the stimulation points defined by labeled nuclei; frames containing significant calcium transients were then used to manually refine each ROI to define the soma of a responsive cell. For each ROI, the change in fluorescence relative to baseline (△F/F) was computed based on a baseline period of 10 frames prior to the onset of any stimulation. Significant calcium transients were defined as events of greater than three standard deviations above the mean for a duration of at least 2 frames. Cells with baseline GCaMP6s signals of more than 2 standard deviations above the mean in a given experiment were excluded from analysis, as GCaMP overexpression can lead to aberrant responsiveness (*Chen et al., 2013*; *Packer et al., 2015*; *Tian et al., 2009*). A synaptic connection was scored if the following criteria were met: (1) a latency of at least 2 ms from stimulus onset, (2) occurring in at least three out of four trials, (3) a jitter of less than 14 ms, (4) a rise time from 10% of peak to 90% of peak of less than 10 ms, and (5) the presence of a calcium transient in the photostimulated cell. Detected connections were confirmed to be excitatory in nature by altering the holding potential of the patched cell to demonstrate a reversal potential of ~0 mV. Connectivity rate was defined as then number of pairs with calcium transients in the presynaptic neuron and postsynaptic currents divided by the number of presynaptic neurons with calcium transients stimulated.

## Acknowledgements

We would like to thank Alipasha Vaziri for advice on temporal focusing, Jason Christie for consultations on implementing temporal focusing on our specific two-photon setup, David Whitney and Dan Wilson for valuable conversations regarding calcium imaging, David Fitzpatrick for comments on the

manuscript, and Asnel Joseph, Nowrin Ahmed, and Laura Conatser for performing stereotaxic AAV injections. GCaMP6s is courtesy of V Jayaraman, R Kerr, D Kim, L Looger, and K Svoboda from the GENIE Project at the Janelia Farm campus of the Howard Hughes Medical Institute. This work was supported by the Max Planck Florida Institute.

## Additional information

### Funding

| Funder | Author |
| --- | --- |
| Max Planck Florida Institute | Christopher A Baker<br>Yishai M Elyada<br>Andres Parra-Martin<br>M McLean Bolton |

The funders had no role in study design, data collection and interpretation, or the decision to submit the work for publication.

### Author contributions

CAB, Conceived the study, Designed and built the optical setup, Engineered DNA constructs and performed temporal focusing experiments, Analyzed data, Wrote the manuscript; YME, Designed and built the optical setup, Drafting or revising the article, Contributed unpublished essential data or reagents; AP, Determined intrinsic electrophysiological properties and conducted paired recording experiments, Analyzed data; MMcLB, Conceived the study, Analyzed data, Wrote the manuscript

### Author ORCIDs

Christopher A Baker, http://orcid.org/0000-0002-0604-8449

### Ethics

Animal experimentation: This study was performed in strict accordance with the recommendations in the Guide for the Care and Use of Laboratory Animals of the National Institutes of Health, and all animals were handled according to protocols approved by the Institutional Animal Care and Use Committee of the Max Planck Florida Institute for Neuroscience.

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
