## [Decision Letter]

[Editors’ note: a previous version of this study was rejected after peer review, but the authors submitted for reconsideration. The first decision letter after peer review is shown below.]

Thank you for choosing to send your work entitled "Single-cell resolution circuit mapping with temporal-focused excitation of soma-targeted channelrhodopsin" for consideration at *eLife*. Your full submission has been evaluated by Eve Marder as the Senior editor and three peer reviewers, one of whom is a member of our Board of Reviewing Editors, and the decision was reached after discussions between the reviewers. Based on our discussions and the individual reviews below, we regret to inform you that the work does not meet the standard for publication in *eLife*.

While the reviewers agreed that the strategy you propose is elegant and potentially very powerful, and could have a major impact on the field of neuroscience, unfortunately they also expressed serious concerns regarding the details of the experiments and the presentation of the results, which dampened enthusiasm for the manuscript. In particular, the section of the manuscript on circuit mapping was felt to be especially weak and would require a major series of additional experiments. As you probably know, *eLife* has a policy of not asking for significant new experiments as part of a revision, thus resulting in the present decision.

*Reviewer #1:*

In this paper, the authors describe an optogenetics method to reach optogenetics activation with single cell resolution. The approach combines the use of temporal focusing with targeted opsins, which restrict the chanelrhodopsin expression to the soma and the apical dendrites. The development of somatic opsins is one of the most promising ways of reaching a true cellular resolution with optogenetics and will surely strongly impact the neuroscience community.

However, in my opinion, the manuscript cannot be published in the present form and requires significant major modifications.

Introduction:

References reporting optical methods using two-photon excitation, diffractive optics and temporal focusing for optogenetics are not extensively cited. Also a state of the art (including eventual references) on alternative approaches to achieve somatic ChR2 targeting will help in appreciating the novelty of the paper. The decision of using Kv2.1 voltage gated potassium channel to confine ChR2 targeting should be discussed more extensively.

Results and Discussion:

Figure 1 (left): The meaning of this figure is not clear. It seems that the authors want here to characterize the optical resolution of the system. If this is the case, lateral resolution will be better characterized by performing a lateral displacement of the excitation spot along the x and y direction and plotting the corresponding curves (similarly as was done in Figure 1, right panel, for axial resolution). The 3D image is very confusing and not necessary.

Figure 2—figure supplement 2: Cross sections through the images are needed in order to appreciate the values for lateral and axial FWHM.

Figure 2: The image showing the Alexa 594 distribution (central bottom panel) has a very reduced fluorescence spreading with respect to the central top one: this difference is not justified as the spreading should be comparable in the two cells: authors should probably choose a better example.

Figure 2: The experiment on acute slice has been done only once: this is not enough to support their conclusion more statistics is needed. They should be able to derive for acute slice a figure similar to Figure 2 and C.

They need to discuss the effect of the planarity of the dendrites in the experiments: the excitation spot has been moved laterally along the objective focal plane, if the dendritic process is axially tilted this could also induce a decrease in the current (more statistic will enables removing this ambiguity).

In order to compare data from non targeted and targeted cells, authors should comment on the time they wait after injections in the two cases. Is this comparable? How long the somatic targeting stays somatic? Is there a critical time window after which the somatic ChR2 starts spreading along dendritic processes? Scale bar should be indicated in the bottom image. Why for the targeted ChR2, data have been taken with larger step?

Figure 2: The authors should better explain how they obtained this figure; how do they define the threshold?

Figure 2: In the caption they write "each position in a map was stimulated with the minimum power that reliably evoked action potential when stimulation was applied to the soma": they should better quantify the meaning of "reliably evoked action potential". Stimulation protocol (pulse duration, pulse frequency) should be indicated in the caption for all the experiments.

Figure 2—figure supplement 1: Not needed.

Figure 3: The data and procedure reported in this figures needed to be better presented and explained.

A picture showing the GCamp6 fluorescence before photostimulation is needed to visualize the distribution of the cells in rest condition.

It is not clear if the cell dye-filled and imaged in A is a ChR2 positive cell. If this is the case, authors need to show the current when the photostimulation spot is placed on the cell. The experiment should be repeated more than a single time to be convincing.

The construct used in Figure 3 uses ChR2 directly linked to GCamp6: this is a very powerful idea and should be better highlighted.

Results section: "[…] owing to the lower efficiency of spike generation by ChR2 in the absence of TF […]" this sentence is wrong. TF does not increase the efficiency of ChR2 excitation but only reduce the out of focus contribution, thus improving axial resolution.

"[…] and a reproducible current with appropriate synaptic delay and kinetics […]" this sentence is very vague, authors should define and quantify what is an "appropriate synaptic delay and kinetics".

The discussion on the biological results of in Figure 3 should be toned down. The paper is a methodological paper with interesting results and does not need in my opinion a biological conclusion that is not supported by enough data.

*Reviewer #2:*

The authors created a new construct that localizes ChR2 to the soma and proximal dendrites of neurons. When combined with two-photon beam shaping methods (e.g. temporal focusing), this should improve the ability to target and stimulate individual densely packed neurons without concurrently activating their neighbors.

While the new construct may alleviate some of the concerns typically associated with optical mapping of connectivity (i.e. the inability to precisely stimulate only neurons of choice), the data presented in this manuscript are far too preliminary to make an impact in the field of circuit mapping.

Detailed comments:

Figure 1: Much more quantification is needed. The important variable for circuit mapping (Figure 3) is whether or not a spike is elicited, rather than the inward current. The authors should determine on what fraction of trials a spike is elicited for each power, for each location. Currently only single-trial raw current-clamp data is shown in Figure 1, but some quantification of this is required, for example:

For the final power chosen, for each neuron, what fraction of trials led to a spike when the spot was directly on the soma, and what fraction of trials led to a spike when the spot was directly, vertically, above the neuron (i.e. position iv), which seems to be the most vulnerable position for eliciting unwanted action potentials?

What was the final power used for the example shown? 61mW is on the threshold of activating the neuron soma directly (position iii), and 89mW (the next power tested) is on the threshold of activating the neuron when the beam is not directed to the soma (position iv).

As far as I can make out, the authors go on to change the protocol later in the paper (Figure 3, "circuit mapping"), using 150ms long pulses in order to generate trains of action potentials. However, all of the analysis in Figure 1 needs to be redone with these experimental parameters, since longer stimulation pulses will increase the probability of unwanted spikes away from the location of light stimulation.

What is the latency to action potential for each of the laser powers?

Figure 2: In panel 2D, the authors should show an example of a "targeted" neuron (i.e. ChR2 localized to the soma), whilst stimulating at points along the apical dendrite at the same density as that shown for the "non-targeted" neuron. Also, the current elicited in the targeted neuron is here lower than the current elicited for the non-targeted neuron, which contradicts panel E, and is not "representative" – what was the stimulation power used in the two cases?

In panel 2G, the interesting variable is the average number of spikes elicited in current clamp and these data would have been more valuable.

Figure 3: The image quality needs to be refined, and some of the somata are poorly defined. This applies particularly to the cells that are assumed to be connected.

The voltage-clamp traces in panel 3C are single trial data. The authors should show multiple traces for each connection to convince the reader that a true connection is present, rather than an EPSC which happens to coincide with light stimulation.

The authors should quantify the calcium signals in all the neurons in the imaged population when a single neuron has been targeted for stimulation (beyond what's shown in Video 1 & 2, which are not informative). Crucially, the authors must show unambiguously that there was only one neuron active on each stimulation trial.

Reviewer #3:

The authors present a novel combination of two known methods, light shaping and opsin targeting, for the purpose of mapping synaptic connections in vitro. This is in principle a very elegant approach for improving the spatial precision of optogenetic activation, currently a key limitation in the field. However, the manuscript has a rather preliminary flavor (several of the key observations appear to be n = 1). The authors are in the position to provide a major advance here by performing a detailed quantification of how accurate and reliable their method is, using ground truth calibrations. For example, the authors have not quantified how accurate their method is with any paired recordings to prove the connections they find are real. They only state that the average connectivity is similar to that in other experiments in which pairs were directly recorded. Most importantly, the lack of detailed quantification (with mean, SD, and N) needs to be addressed prior to publication.

Major comments:

1) There are major details missing in Figure 1. What is the mean action potential reliability and resolution, i.e. the grand average result of Figure 1 across all neurons? What powers were typically used for AP generation at the soma in these experimental conditions? What are the max currents observed? Please provide mean, SD, and N. Note that the figure was not created with the construct that was ultimately used, which is a weakness.

2) How many cells were used to generate the data in Figure 2—figure supplement 2? It appears that some of the differences are statistically borderline and without complete data including the sample size it is difficult to determine the reliability of this result. Also, how did the authors determine the number of significant digits to include?

3) In describing Figure 3, the authors mention that 3 photostimulation trials are performed at each location. Could the authors please show raw trials, perhaps in a lighter shade behind the average, to indicate the reliability of observed connections?

4) Figure 3 uses a different stimulation duration that rest of paper – the photostimulation time has been increased to 150 ms for Figure 3. As this value doesn't match the previous calibrations, it is very difficult to use the data in Figure 2 to calibrate Figure 3. How does the longer duration affect spatial resolution, action potential threshold, etc.?

5) Many of the calcium imaging transients in Figure 3 are quite large, and sometimes double-peaked when there is only one EPSC observed (Figure 3, bottom row, red square). How do the authors explain the discrepancy between the fact that these long photostimulations (150 ms!) may very well induce more than one action potential, but only one post-synaptic response is observed? Many cortical synapses may depress, but not sufficiently to explain these observations.

6) What is the cutoff for a connection and how reliable is this? For example, in the bottom row of Figure 3, fourth from the right, there is a large calcium transient and some tiny EPSCs – could these be a weak connection?

7) How often do the authors observe failures to confirm pre-synaptic action potential generation with imaging? They only say "occasionally". Excluding these from analysis could heavily bias estimation of connectivity rates!

8) In Figure 3, the authors photostimulate 192 different locations in a grid-like fashion. They don't aim to zap neurons directly, but rather by shooting at many locations, they hope to hit some neurons by chance. A quick segmentation of the image to find neurons and shoot them directly would improve accuracy, reliability, and potentially even be more efficient! Why do the authors not target neurons initially?

9) Have the authors repeated the experiment shown in Figure 3 more than once? If so, please present some grand average data.

10) Losonczy et al. 2010, cited by the authors, shows effective activation of axons. How can the authors be sure that is not occurring here? Can they provide some presynaptic patch confirmations of any of the connections they see?

[Editors’ note: what now follows is the decision letter after the authors submitted for further consideration.]

Thank you for submitting your article "Single-cell resolution circuit mapping with temporal-focused excitation of soma-targeted channelrhodopsin" for consideration by *eLife*. Your article has been reviewed by two peer reviewers, and the evaluation has been overseen by a Reviewing Editor and Eve Marder as the Senior Editor.

The reviewers have discussed the reviews with one another and the Reviewing Editor has drafted this decision to help you prepare a revised submission. We hope you will be able to submit the revised version within two months, so please let us know if you have any questions first.

Summary:

The development of somatically targeted opsins is one of the most promising ways of achieving true cellular resolution with optogenetics and will surely have a strong impact the neuroscience community. The authors have considerably improved the quality of the manuscript by adding more data and statistical analysis. There is a strong case now for arguing that this new construct is better suited for mapping connectivity in the circuit using targeted optogenetic stimulation. However a few points of the paper still need additional work before the manuscript is ready to proceed towards publication. We encourage the authors to proceed with these final experiments as a matter of urgency, as this is a highly competitive field.

Essential revisions:

1) The power levels used in the different experiments are often missing, and this information is crucial to appreciate the spatial resolution achieved in the experiments (e.g. how far are the powers used from the saturation value?). The powers used to evoke a single AP are rather high and latency and jittering are rather long compared to what has been reported in the literature. This point is particularly weak considering that in several parts of the manuscript the authors insist on the "enhanced sensitivity "of the targeted opsin. Overall this implies that the opsins (somatic or not) used in this experiments are not very efficient and may not be suitable for experiments requiring e.g. multi-spot stimulation. Many datapoints e.g. the ones showing cellular resolution, or the connectivity experiments, are only performed using the targeted opsins and it is difficult to appreciate their importance if one can't compare the same experiments performed with the non-targeted version.

2) In order to showcase the advantage of the 'new targeted' construct, it is crucial to include the axial and lateral profile of spike probability also for the 'non-targeted' construct in Figure 2. Please add this quantification to existing panels in this figure. State the power at which these curves were obtained.

3) Figure 1: line scans to demonstrate somatic targeting are all done along dendritic processes, while no information or data are provided to show the expression confinement along axons. This would be helpful.

4) Figure 1: "each pixel in the map show the direct current" are the authors plotting the peak current here? Moreover from this map it is difficult to understand the depolarization achieved. The same experiment performed in current clamp would allow us to learn about the spike probability for spot placed out of the target, which ultimately is the key elements to support the necessity of the somatic opsin for the connectivity experiments in Figure 3 or to appreciate the enhanced spatial resolution (see next comment).

5) Figure 3 is nice. The quantification of these connectivity mapping experiments could be included in this figure rather than only in the Results section of the text. For completion, please add an additional example of another such slice experiment in an extra supplementary figure. Also, the information on the stimulation protocol used here is very vague: "each cell was stimulated in series with 2 seconds between stimuli": how many stimuli? What power did they use? How confined is the response if experiments as the ones showed in Figure 1 are done using this protocol? How do these results compare if similar experiments are done with a non-somatic opsin?

6) In the discussion the authors justify the use of high power and long photostimulation power:

"We did not take full advantage of the temporal precision capability of TF to fire action potentials in our current study, instead focusing on a screening method that would identify connections without optimizing the amount of power that would fire each potential presynaptic neuron with minimal latency. We therefore chose longer pulses at a power sufficient to fire most neurons and generate trains of action potentials, which would elicit stronger signals with calcium indicators. For experiments requiring temporal precision, the minimization of action potential latency requires optimization of excitation area and laser power"

This paper should convince us about the use of a new optogenetic construct, and (as discussed above) a more detailed characterization of the opsin showing the photostimulation area and laser power that enables AP generation with a temporal resolution and precision comparable to what has been achieved in the literature is important and should be carried out.

7) The sentence "Furthermore, these techniques could also be used in vivo, where the enhanced sensitivity of the targeted ChR2 makes it especially attractive" is misleading: in the paper the authors do show that the targeted version is more sensitive than the non-targeted one. But in both cases they use excitation powers much higher than what has been achieved in the literature and demonstrate performances (temporal resolution, latency and jittering) inferior to what has been achieved with ChR2 or C1V1 by other labs. This should be reworded.

[Editors' note: further revisions were requested prior to acceptance, as described below.]

Thank you for resubmitting your work entitled "Single-cell resolution circuit mapping in mouse brain with temporal-focused excitation of soma-targeted channelrhodopsin" for further consideration at *eLife*. Your revised article has been favorably evaluated by Eve Marder as the Senior editor, a Reviewing editor, and two reviewers.

The manuscript has been improved but there are some remaining issues that need to be addressed before acceptance, as outlined below:

1) There remain concerns about the intensity and duration of the illumination pulses used (150 ms). This suggests that the construct is not very efficient. Why were such long pulses used? Were shorter pulses used in some experiments? The authors should either demonstrate that their construct is also effective in triggering spikes when using shorter pulses, or provide a convincing justification for the use of longer pulses.

2) Please add to the Methods section some of the text that is currently a response to point 1, related to stimulation power needed to excite the cells. ("Our manuscript reports excitation powers of between 15 and 285 mW (for both types of opsins); assuming our spot size to be at least 10 μm wide and 10 μm thick, we are using powers of no greater than 0.2 to 3.8 mW/μm^2^). Expressing power as mW/μm^2^ (rather than incident power in hundreds of mW) will be useful for readers.

3) 'Single-cell resolution' is advertised in the title, but is not well supported. We suggest changing the beginning of the title to 'Cellular resolution…'.

4) Please extend the comparison between your results and those of Wu et al. 2013 Plos ONE (since they originated the somatic restriction strategy).

---

## [Author Response]

[Editors’ note: the author responses to the first round of peer review follow.]

*Reviewer #1:*

*Introduction:*

*References reporting optical methods using two photon excitation, diffractive optics and temporal focusing for optogenetics are not extensively cited. Also a state of the art (including eventual references) on alternative approaches to achieve somatic ChR2 targeting will help in appreciating the novelty of the paper. The decision of using Kv2.1 voltage gated potassium channel to confine ChR2 targeting should be discussed more extensively.*

We have lengthened the Introduction to provide a larger context for the mapping experiments including additional citations for temporal focusing and two-photon microscopy. A separate paragraph in the Discussion has been included that provides a review of other approaches that have been taken to achieve somatic targeting, and the advantages of using Kv2.1 to achieve opsin subcellular restriction.

*Results and Discussion:*

*Figure 1 (left): The meaning of this figure is not clear. It seems that the authors want here to characterize the optical resolution of the system. If this is the case, lateral resolution will be better characterized by performing a lateral displacement of the excitation spot along the x and y direction and plotting the corresponding curves (similarly as was done in Figure 1, right panel, for axial resolution). The 3D image is very confusing and not necessary.*

Indeed, the reviewer understood our original aim and provided a better alternative to achieving this end. The original figure referenced has been omitted and replaced with lateral and axial resolution measurements in Figure 2.

*Figure 2—figure supplement 2: Cross sections through the images are needed in order to appreciate the values for lateral and axial FWHM.*

Fluorescence measurements have been added, now as Figure 1—figure supplement 2. Fitting Gaussian functions to the measurements revealed the full width at half maximum to be even smaller than previously estimated.

*Figure 2: The image showing the Alexa 594 distribution (central bottom panel) has a very reduced fluorescence spreading with respect to the central top one: this difference is not justified as the spreading should be comparable in the two cells: authors should probably choose a better example.*

Two different neurons have been chosen for the representation.

*Figure 2: The experiment on acute slice has been done only once: this is not enough to support their conclusion more statistics is needed. They should be able to derive for acute slice a figure similar to Figure 2 and C.*

Experiments on acute slices were repeated for 10 cells expressing untargeted ChR2 and 13 cells expressing targeted ChR2. Representative data are presented in Figure 1, with averages in Figure 1. The data from cultured neurons have been dropped to make room for the new slice experiments.

*They need to discuss the effect of the planarity of the dendrites in the experiments: the excitation spot has been moved laterally along the objective focal plane, if the dendritic process is axially tilted this could also induce a decrease in the current (more statistic will enables removing this ambiguity).*

We have tried to clarify that each dendrite under study was followed carefully as its depth changed within the slice, and that stimulation spots were chosen such that the temporal focusing spot contained a complete stretch (>10μm) of dendrite within a single focal plane. In addition, our inclusion of more cells should average out any remaining effects of dendrite planarity, as the reviewer suggests.

*In order to compare data from non targeted and targeted cells, authors should comment on the time they wait after injections in the two cases. Is this comparable? How long the somatic targeting stays somatic? Is there a critical time window after which the somatic ChR2 starts spreading along dendritic processes?*

This information was not as explicit or extensive in the original version of the manuscript. It now appears in the Materials and methods section. All preparations were examined between 3 and 4 weeks post-injection. Somatic restriction has been observed at 5 weeks after injection, but we have not examined later time points.

*Scale bar should be indicated in the bottom image. Why for the targeted ChR2, data have been taken with larger step?*

As described above, the exact positions along dendrites were chosen in part based on planarity.For the representative example (Figure 1), we tried to choose a pair of neurons where the stimulation locations were similarly spaced.

*Figure 2: The authors should better explain how they obtained this figure; how they define the threshold?*

The threshold is defined as the minimum intensity that produced action potentials in ten out of ten consecutive trials. This is now explained at multiple points within the text.

Figure 2: In the caption they write "each position in a map was stimulated with the minimum power that reliably evoked action potential when stimulation was applied to the soma": they should better quantify the meaning of "reliably evoked action potential".

This is the aforementioned threshold intensity for a given neuron.

*Stimulation protocol (pulse duration, pulse frequency) should be indicated in the caption for all the experiments.*

The captions for Figure 1–Figure 3 now contain descriptions of the stimulation protocol.

Figure 2—figure supplement 1: Not needed.

*Figure 3: The data and procedure reported in this figures needed to be better presented and explained.*

*A picture showing the GCamp6 fluorescence before photostimulation is needed to visualize the distribution of the cells in rest condition.*

This figure has been completely redone in the new manuscript. Baseline GCaMP fluorescence is shown in Figure 3.

*It is not clear if the cell dye-filled and imaged in A is a ChR2 positive cell. If this is the case, authors need to show the current when the photostimulation spot is placed on the cell. The experiment should be repeated more than a single time to be convincing.*

In the original manuscript, the cell shown was indeed ChR2-positive. In all our mapping experiments, the patched cells express ChR2 and we use their responsiveness to calibrate the photostimulation power. We have included examples of current traces found with direct stimulation of neuronal somata in Figure 1. As for repetition of the experiment, we have indeed done this and report average statistics in the text of the Results section (seventh paragraph).

*The construct used in Figure 3 uses ChR2 directly linked to GCamp6, this is a very powerful idea and should be better highlighted.*

We agree with the reviewer that this is a potentially powerful approach, however, in practice we found that this method actually led to decreased GCaMP responsiveness that contributed to poor signal to noise ratios and suboptimal image quality. We have not completely characterized the reasons for this but suspect that linking high opsin expression to high GCaMP expression on the same construct leads to many cells with GCaMP in their nuclei that no longer exhibit calcium transients. As a result, we turned to the approaches that other groups employ—using a dilution of the GCaMP virus—and combined this with nuclear labeling of neurons making identification easier for stimulation by temporal focusing.

*Results section: "[…] owing to the lower efficiency of spike generation by ChR2 in the absence of TF […]" this sentence is wrong. TF does not increase the efficiency of ChR2 excitation but only reduce the out of focus contribution, thus improving axial resolution.*

We thank the reviewer for this correction. The efficiency of ChR2 excitation is quite high as indicated by its two-photon cross section. We intended to refer to the low single-channel conductance of ChR2 and suggest that when the excitation is axially confined, a rapid scanning approach or a light sculpting technique is needed to quickly activate many ChR2 molecules and drive spike generation. We have clarified this in the text and actually expanded the portion of the Results that discusses ChR2 activation by the imaging beam (fourth paragraph).

*"[…] and a reproducible current with appropriate synaptic delay and kinetics […]" this sentence is very vague, authors should define and quantify what is an "appropriate synaptic delay and kinetics"*

We define a bona fide postsynaptic response as exhibiting a rise time of less than 10 ms, and having a greater than 2 ms latency from the photostimulation and with less than 14 ms jitter.

These criteria are now described in the Materials and methods section (subsection “Calcium imaging and resolution studies”).

*The discussion on the biological results of in Figure 3 should be toned down. The paper is a methodological paper with interesting results and does not need in my opinion a biological conclusion that is not supported by enough data.*

We agree, and do not desire to make a biological statement about layer 3-local excitatory connectivity. Rather, we think it is important to provide the reader with a measure of the sensitivity and reliability of this new approach for mapping synaptic connections with the current gold standard: paired patch recording. We have revised the text to clarify the rationale for inclusion of this comparison.

*Reviewer #2:*

*[…] Figure 1: Much more quantification is needed. The important variable for circuit mapping (Figure 3) is whether or not a spike is elicited, rather than the inward current. The authors should determine on what fraction of trials a spike is elicited for each power, for each location. Currently only single-trial raw current-clamp data is shown in Figure 1, but some quantification of this is required, for example:*

*For the final power chosen, for each neuron, what fraction of trials led to a spike when the spot was directly on the soma, and what fraction of trials led to a spike when the spot was directly, vertically, above the neuron (i.e. position iv), which seems to be the most vulnerable position for eliciting unwanted action potentials?*

*What was the final power used for the example shown? 61mW is on the threshold of activating the neuron soma directly (position iii), and 89mW (the next power tested) is on the threshold of activating the neuron when the beam is not directed to the soma (position iv).*

In response to the reviewer’s concerns we have abandoned the original figure in favor of an analysis examining spike probability for ten neurons over ten trials at axially and laterally displaced locations. These data are now reported in the current manuscript as Figure 2.

*As far as I can make out, the authors go on to change the protocol later in the paper (Figure 3, "circuit mapping"), using 150ms long pulses in order to generate trains of action potentials. However, all of the analysis in Figure 1 needs to be redone with these experimental parameters, since longer stimulation pulses will increase the probability of unwanted spikes away from the location of light stimulation.*

All experiments in the paper have now been performed with the longer 150 ms stimulation protocol.

*What is the latency to action potential for each of the laser powers?*

The average latency for the mapping experiments was 39 ms, which is now mentioned in the text.

*Figure 2: In panel 2D, the authors should show an example of a "targeted" neuron (i.e. ChR2 localized to the soma), whilst stimulating at points along the apical dendrite at the same density as that shown for the "non-targeted" neuron. Also, the current elicited in the targeted neuron is here lower than the current elicited for the non-targeted neuron, which contradicts panel E, and is not "representative" – what was the stimulation power used in the two cases?*

The new version of this experiment is presented in Figure 1. The stimulation density is similar for both the targeted and nontargeted examples, and examining over 10 cells per group allowed us to choose truly representative cells that showcase the heightened current achieved with the targeted construct even with equivalent power (167mW in these cases).

*In panel 2G, the interesting variable is the average number of spikes elicited in current clamp and these data would have been more valuable.*

As mentioned previously, spike probability as a function of lateral or axial displacement is now included in Figure 2. For our mapping experiments evaluating small synaptic currents in response to stimulation of putative presynaptic cells, however, showing a reduction in the currents evoked by direct stimulation of the dendritic arbor of the patched cell is still relevant.

*Figure 3: The image quality needs to be refined, and some of the somata are poorly defined. This applies particularly to the cells that are assumed to be connected.*

New images of cells are provided in Figure 3 fluorescent nuclear label and better GCaMP expression allowed for more defined images of all cells.

*The voltage-clamp traces in panel 3C are single trial data. The authors should show multiple traces for each connection to convince the reader that a true connection is present, rather than an EPSC which happens to coincide with light stimulation.*

All four voltage clamp traces used to determine the presence or absence of a synaptic connection are now shown in Figure 3.

*The authors should quantify the calcium signals in all the neurons in the imaged population when a single neuron has been targeted for stimulation (beyond what's shown in Video 1 & 2, which are not informative). Crucially, the authors must show unambiguously that there was only one neuron active on each stimulation trial.*

These data are now included in Figure 3, where the calcium signal is shown for each targeted neuron in a given experiment. Every connection is unequivocally associated with only one active neuron.

*Reviewer #3:*

*[…] 1) There are major details missing in Figure 1. What is the mean action potential reliability and resolution, i.e. the grand average result of Figure 1 across all neurons? What powers were typically used for AP generation at the soma in these experimental conditions? What are the max currents observed? Please provide mean, SD, and N. Note that the figure was not created with the construct that was ultimately used, which is a weakness.*

The resolution of spike generation in 10 cells expressing the targeted construct is now presented in Figure 2. These measurements were determined at the threshold power for each particular cell, the distribution of which is shown in Figure 2. The average currents at each power are shown in Figure 2. There was no difference between targeted and untargeted cells in the average current necessary to evoke action potentials (i.e., rheobase), only a difference in the amount of stimulation power required to evoke such current.

*2) How many cells were used to generate the data in Figure 2—figure supplement 2? It appears that some of the differences are statistically borderline and without complete data including the sample size it is difficult to determine the reliability of this result. Also, how did the authors determine the number of significant digits to include?*

These data are now in Figure 2—figure supplement 2, and include N for all measurements. Significant digits are determined by the measurement limits of the Clampex patch clamp software.

*3) In describing Figure 3, the authors mention that 3 photostimulation trials are performed at each location. Could the authors please show raw trials, perhaps in a lighter shade behind the average, to indicate the reliability of observed connections?*

All raw trials are now shown in the new Figure 3, with calcium traces overlaid in a lighter shade and electrophysiological traces displayed adjacent to each other.

*4) Figure 3 uses a different stimulation duration that rest of paper – the photostimulation time has been increased to 150 ms for Figure 3. As this value doesn't match the previous calibrations, it is very difficult to use the data in Figure 2 to calibrate Figure 3. How does the longer duration affect spatial resolution, action potential threshold, etc.?*

All data in the paper are now derived from 150ms stimulation.

*5) Many of the calcium imaging transients in Figure 3 are quite large, and sometimes double-peaked when there is only one EPSC observed (Figure 3, bottom row, red square). How do the authors explain the discrepancy between the fact that these long photostimulations (150 ms!) may very well induce more than one action potential, but only one post-synaptic response is observed? Many cortical synapses may depress, but not sufficiently to explain these observations.*

Indeed, the longer stimulation protocol was used in hopes of generating trains of presynaptic action potentials. While single postsynaptic currents are observed (some neurons only generate one action potential regardless of stimulation intensity), we frequently saw large calcium transients and trains of postsynaptic currents, which are shown in the new Figure 3.

*6) What is the cutoff for a connection and how reliable is this? For example, in the bottom row of Figure 3, fourth from the right, there is a large calcium transient and some tiny EPSCs – could these be a weak connection?*

In the original version of the manuscript, the currents described by the reviewer did not occur across multiple trials. In the new experiments reported here, we require that events occur in at least three out of four trials and with less than 14 ms jitter and a greater than 2 ms latency from the photostimulation. These criteria are now described in the Materials and methods section (subsection “Calcium imaging and connection analysis”).

*7) How often do the authors observe failures to confirm pre-synaptic action potential generation with imaging? They only say "occasionally". Excluding these from analysis could heavily bias estimation of connectivity rates!*

The exact numbers of excluded connections from the current study are now described in the Results. For the majority of cells (5 out of 7 in the current study), every connection found could be associated with a somatic calcium transient. Out of the 36 connections identified across all cells in the present study, 3 were excluded due to lack of a somatic calcium signal.

*8) In Figure 3, the authors photostimulate 192 different locations in a grid-like fashion. They don't aim to zap neurons directly, but rather by shooting at many locations, they hope to hit some neurons by chance. A quick segmentation of the image to find neurons and shoot them directly would improve accuracy, reliability, and potentially even be more efficient! Why do the authors not target neurons initially?*

We originally hoped that baseline GCaMP fluorescence would be sufficient to target neurons as the reviewer suggests, however, we found that many neurons with significant calcium responses to TF stimulation were impossible to identify under baseline imaging conditions. We agree with the reviewer that segmentation would be preferable, and developed a new construct in which neuronal nuclei were fluorescently labeled for easy targeting for TF stimulation. All mapping studies in the current version of the paper use this construct.

*9) Have the authors repeated the experiment shown in Figure 3 more than once? If so, please present some grand average data.*

The data for seven different mapping experiments are now described in the Results section (seventh paragraph).

*10) Losonczy et al. 2010, cited by the authors, shows effective activation of axons. How can the authors be sure that is not occurring here? Can they provide some presynaptic patch confirmations of any of the connections they see?*

Targeted ChR2 expression is not seen in the axons of dye-filled cultured neurons, which is consistent with the known exclusion of Kv2.1 from axons beyond the initial segment.

[Editors’ note: the author responses to the re-review follow.]

*Essential revisions:*

*1) The power levels used in the different experiments are often missing, and this information is crucial to appreciate the spatial resolution achieved in the experiments (e.g. how far are the powers used from the saturation value?). The powers used to evoke a single AP are rather high and latency and jittering are rather long compared to what has been reported in the literature. This point is particularly weak considering that in several parts of the manuscript the authors insist on the "enhanced sensitivity "of the targeted opsin. Overall this implies that the opsins (somatic or not) used in this experiments are not very efficient and may not be suitable for experiments requiring e.g. multi-spot stimulation. Many datapoints e.g. the ones showing cellular resolution, or the connectivity experiments, are only performed using the targeted opsins and it is difficult to appreciate their importance if one can't compare the same experiments performed with the non-targeted version.*

The power levels used in most experiments were calibrated for each cell to generate a single action potential; we chose this approach to compare resolution and sensitivity across multiple cells that may differ in absolute levels of opsin expression. We have made an extra effort to explicitly mention this rationale as well as the average powers chosen for each experiment; for sample mapping experiments in Figure 3 and Figure 4, we have also included specific power levels in the figure legends.

With temporal focusing excitation distributed across a volume, the power levels for opsin excitation are expected to be higher than those used when scanning a much smaller diffraction-limited spot. In earlier papers using temporal focusing stimulation of ChR2, Andrasfalvy et al. (*PNAS,* 2010) reported action potential generation anywhere with 300-400 mW power at the sample plane, and Losonczy et al (Nature Neurosci., 2010) report power levels between 50 and 500 mW. Our manuscript reports excitation powers of between 15 and 285 mW (for both types of opsins); assuming our spot size to be at least 10μm wide and 10μm thick, we are using powers of no greater than 0.2 to 3.8 mW/μm^2^. This approaches but does not surpass excitation powers of 0.45 mW/μm^2^ reported by the Emiliani group, but differences in opsin delivery and expression may account for some of the discrepancy. Finally, Rickgauer et al. (Nature Neurosci., 2014) were able to use less than 100 mW over the entire temporal focusing disc when employing C1V1, which could be due to the enhanced optical response of that opsin relative to ChR2.

We agree with the reviewers that additional comparisons between nontargeted and targeted opsins would reveal the utility of a somatic targeting approach. We have taken this approach to heart and included new comparisons in Figure 2, and also added a new Figure 2—figure supplement 3 and Figure 4 (see below).

*2) In order to showcase the advantage of the 'new targeted' construct, it is crucial to include the axial and lateral profile of spike probability also for the 'non-targeted' construct in Figure 2. Please add this quantification to existing panels in this figure. State the power at which these curves were obtained.*

The manuscript now has a new version of Figure 2 with the nontargeted resolutions incorporated (see also sixth paragraph of Results). As alluded to above, the power levels for all the resolution experiments (including both Figure 1 and Figure 2) vary from cell to cell and are set to be the minimum power that generates a single action potential in ten out of ten trials. The average powers for these experiments are reported in third paragraph of Results and a mistaken power level was corrected.

*3) Figure 1: line scans to demonstrate somatic targeting are all done along dendritic processes, while no information or data are provided to show the expression confinement along axons. This would be helpful.*

We conducted similar optical stimulation experiments on axonal processes, added more detailed imaging, and included the results as Figure 1—figure supplement 3 and in the Results (fourth paragraph).

*4) Figure 1: "each pixel in the map show the direct current" are the authors plotting the peak current here? Moreover from this map it is difficult to understand the depolarization achieved. The same experiment performed in current clamp would allow us to learn about the spike probability for spot placed out of the target, which ultimately is the key elements to support the necessity of the somatic opsin for the connectivity experiments in Figure 3 or to appreciate the enhanced spatial resolution (see next comment).*

Indeed, we are plotting the peak current amplitudes in Figure 1. This voltage clamp experiment remains an important demonstration of the extent of direct currents that overwhelm smaller synaptic events in mapping experiments, which can be seen now in the new Figure 4. We agree that the spike probability at off target locations is worthwhile information and were able to investigate this in the context of actual connectivity mapping experiments by extracting information from our previous data as well as new experiments with nontargeted opsin. When using a nontargeted construct, we found an increased probability of calcium transient generation in off-target cells when stimulating separate cells in the slice. Moreover, the average distance between an off-target activated cell and the stimulation point was greater when using nontargeted ChR2. These observations are now reported in the Results.

*5) Figure 3 is nice. The quantification of these connectivity mapping experiments could be included in this figure rather than only in the Results section of the text. For completion, please add an additional example of another such slice experiment in an extra supplementary figure. Also, the information on the stimulation protocol used here is very vague: "each cell was stimulated in series with 2 seconds between stimuli": how many stimuli? What power did they use? How confined is the response if experiments as the ones showed in Figure 1 are done using this protocol? How do these results compare if similar experiments are done with a non-somatic opsin?*

We appreciate the reviewer’s comments about the new figure, and have incorporated more quantification from multiple connectivity experiments into the figure legend, as requested. We have also included another example slice experiment as Figure 3—figure supplement 1. We have also tried to explain in the figure legend that each cell is stimulated sequentially at the onset of every other imaging frame (i.e., 2.4 seconds after the previous cell in the field). After going through all the cells in the field (40-80 cells), the entire protocol is repeated for three additional iterations to produce the results in the figures. We have reported the average power value across the mapping experiments seventh paragraph of Results and the specific power levels for each representative experiment within the figure legend. Finally, we appreciate the suggestion to perform mapping experiments with nontargeted ChR2, and have included an example as Figure 4. This allowed us to demonstrate unintended activation of off-target cells as well as the extent to which direct activation of the dendritic arbor occurs and can make interrogation of local connectivity problematic.

*6) In the discussion the authors justify the use of high power and long photostimulation power:*

*"We did not take full advantage of the temporal precision capability of TF to fire action potentials in our current study, instead focusing on a screening method that would identify connections without optimizing the amount of power that would fire each potential presynaptic neuron with minimal latency. […] For experiments requiring temporal precision, the minimization of action potential latency requires optimization of excitation area and laser power"*

*This paper should convince us about the use of a new optogenetic construct, and (as discussed above) a more detailed characterization of the opsin showing the photostimulation area and laser power that enables AP generation with a temporal resolution and precision comparable to what has been achieved in the literature is important and should be carried out.*

Our goal with this manuscript was to provide a baseline system from which to base further refinements of optogenetic mapping experiments and to establish the utility of somatic opsin targeting for such an approach. We therefore focused on greater characterization of spatial resolution of the targeted ChR2 and less on temporal precision. Nevertheless, we now include data that show action potential latency consistent with that achieved in the literature can be obtained by increasing stimulation power (Figure 2—figure supplement 3). Our mapping experiments in Figure 3 and Figure 4 also demonstrate that the advantages in spatial resolution with the targeted construct are maintained at higher stimulation powers that produce shorter latencies.

*7) The sentence "Furthermore, these techniques could also be used in vivo, where the enhanced sensitivity of the targeted ChR2 makes it especially attractive" is misleading: in the paper the authors do show that the targeted version is more sensitive than the non-targeted one. But in both cases they use excitation powers much higher than what has been achieved in the literature and demonstrate performances (temporal resolution, latency and jittering) inferior to what has been achieved with ChR2 or C1V1 by other labs. This should be reworded.*

We altered the statement to focus more on spatial resolution (final paragraph of Discussion).

*[Editors' note: further revisions were requested prior to acceptance, as described below.]*

*The manuscript has been improved but there are some remaining issues that need to be addressed before acceptance, as outlined below:*

*1) There remain concerns about the intensity and duration of the illumination pulses used (150 ms). This suggests that the construct is not very efficient. Why were such long pulses used? Were shorter pulses used in some experiments? The authors should either demonstrate that their construct is also effective in triggering spikes when using shorter pulses, or provide a convincing justification for the use of longer pulses.*

We implemented a longer stimulation protocol as this provided the greatest probability of eliciting trains of action potentials, leading to increased calcium signals in mapping experiments. In an earlier version of our manuscript, we characterized responses for both types of opsins using a shorter stimulation time (32 ms). We have now included some of these data as an additional figure supplement (Figure 2—figure supplement 3) and added appropriate commentary in the text (fourth paragraph of Results). Moreover, the ability to reduce latency to less than 10 ms with increased power (Figure 2—figure supplement 2) suggests that even shorter stimulation pulses may still be effective.

*2) Please add to the Methods section some of the text that is currently a response to point 1, related to stimulation power needed to excite the cells. ("Our manuscript reports excitation powers of between 15 and 285 mW (for both types of opsins); assuming our spot size to be at least 10 μm wide and 10 μm thick, we are using powers of no greater than 0.2 to 3.8 mW/μm^2^). Expressing power as mW/*μ*m^2^ (rather than incident power in hundreds of mW) will be useful for readers.*

We have now expressed all powers in the text and figures as mW/μm^2^, and added an explanatory note to the Materials and methods section (subsection “Stimulation and resolution studies”).

*3) 'Single-cell resolution' is advertised in the title, but is not well supported. We suggest changing the beginning of the title to 'Cellular resolution…'.*

We have made the requested change.

*4) Please extend the comparison between your results and those of Wu et al. 2013 Plos ONE (since they originated the somatic restriction strategy).*

We extended the discussion of the Wu paper, mentioning their demonstration of somatic restriction by physiologic criteria (second paragraph of Discussion). As they measured the physiologic responses of cells under very different conditions than in our study, it is difficult to directly compare their estimates of opsin restriction with our own.